# Regadenoson for the treatment of COVID-19: A five case clinical series and mouse studies

Joseph Rabin[1], Yunge Zhao[1], Ezzat Mostafa[1], Manal Al-Suqi[1], Emily Fleischmann[1], Mark R. Conaway[2], Barbara J. Mann[3], Preeti Chhabra[4], Kenneth L. Brayman[4], Alexander Krupnick[1], Joel Linden[1,3]*, Christine L. Lau[1]*

1 Department of Surgery, Division of Thoracic, University of Maryland, Baltimore, Maryland, United States of America, 2 Department of Public Health Sciences, University of Virginia, Charlottesville, Virginia, United States of America, 3 Department of Medicine, Division of Infectious Diseases and International Health, University of Virginia, Charlottesville, Virginia, United States of America, 4 Department of Surgery, University of Virginia, Charlottesville, Virginia, United States of America

⊕ These authors contributed equally to this work.

* cllau@som.umaryland.edu (CLL); jl4v@virginia.edu (JL)

## Abstract

### Background

Adenosine inhibits the activation of most immune cells and platelets. Selective adenosine A2A receptor (A2AR) agonists such as regadenoson (RA) reduce inflammation in most tissues, including lungs injured by hypoxia, ischemia, transplantation, or sickle cell anemia, principally by suppressing the activation of invariant natural killer T (iNKT) cells. The anti-inflammatory effects of RA are magnified in injured tissues due to induction in immune cells of A2ARs and ecto-enzymes CD39 and CD73 that convert ATP to adenosine in the extracellular space. Here we describe the results of a five patient study designed to evaluate RA safety and to seek evidence of reduced cytokine storm in hospitalized COVID-19 patients.

### Methods and findings

Five COVID-19 patients requiring supplemental oxygen but not intubation (WHO stages 4–5) were infused IV with a loading RA dose of 5 µg/kg/h for 0.5 h followed by a maintenance dose of 1.44 µg/kg/h for 6 hours, Vital signs and arterial oxygen saturation were recorded, and blood samples were collected before, during and after RA infusion for analysis of CRP, D-dimer, circulating iNKT cell activation state and plasma levels of 13 proinflammatory cytokines. RA was devoid of serious side effects, and within 24 hours from the start of infusion was associated with increased oxygen saturation ($93.8 \pm 0.58$ vs $96.6 \pm 1.08\%$, $P<0.05$), decreased D-dimer ($754 \pm 17$ vs $518 \pm 98$ ng/ml, $P<0.05$), and a trend toward decreased CRP ($3.80 \pm 1.40$ vs $1.98 \pm 0.74$ mg/dL, $P = 0.075$). Circulating iNKT cells, but not conventional T cells, were highly activated in COVID-19 patients (65% vs 5% CD69+). RA infusion for 30 minutes reduced iNKT cell activation by 50% ($P<0.01$). RA infusion for 30 minutes did not influence plasma cytokines, but infusion for 4.5 or 24 hours reduced levels of 11 of 13 proinflammatory cytokines. In separate mouse studies, subcutaneous RA infusion from Alzet minipumps at 1.44 µg/kg/h increased 10-day survival of SARS-CoV-2-infected K18-hACE2 mice from 10 to 40% ($P<0.001$).

**Data Availability Statement:** All relevant data are within the manuscript, its Supporting Information files and the supplemental data files.

**Funding:** This study is supported by the following funding sources: Specific grant number: Award Number: R01HL128492-05S1 Initials of authors who received the award: CLL Full names of commercial companies that funded the study: NHLBI Division of Intramural Research Initials of authors who received salary: YZ, EM, MA, MRC, BJM, JL, CLL URLs to sponsors' websites: https://www.nhlbi.nih.gov/about/divisions/division-intramural-research Specific grant number: Award Number: MC10 Initials of authors who received the award: JL Full names of commercial companies that funded the study: Manning Fund Initials of authors who received salary: JL, BJM, KLB, PC URLs to sponsors' websites: Manning Family Foundation (themanningfamilyfoundation.org) Specific grant number: Award Number: ISR005923 Initials of authors who received the award: CLL Full names of commercial companies that funded the study: Astellas Pharma US Initials of authors who received salary: This funding did not pay any author any salary URLs to sponsors' websites: Astellas Pharma US | Changing Tomorrow | Home Specific grant number: Award Number: R01AI145108 Initials of authors who received the award: AK Full names of commercial companies that funded the study: Division of Microbiology and Infectious Diseases, National Institute of Allergy and Infectious Diseases Initials of authors who received salary: AK URLs to sponsors' websites: Division of Microbiology and Infectious Diseases | NIH: National Institute of Allergy and Infectious Diseases Specific grant number: Award Number: P01 AI116501 Initials of authors who received the award: AK Full names of commercial companies that funded the study: Division of Microbiology and Infectious Diseases, National Institute of Allergy and Infectious Diseases Initials of authors who received salary: AK URLs to sponsors' websites: Division of Microbiology and Infectious Diseases | NIH: National Institute of Allergy and Infectious Diseases We affirmed that the above sponsors or funders played NO role in: Study design, Data collection and analysis, Decision to publish, Preparation of the manuscript of this study.

**Competing interests:** The authors have declared that no competing interests exist.

## Conclusions

Infused RA is safe and produces rapid anti-inflammatory effects mediated by A2A adenosine receptors on iNKT cells and possibly in part by A2ARs on other immune cells and platelets. We speculate that iNKT cells are activated by release of injury-induced glycolipid antigens and/or alarmins such as IL-33 derived from virally infected type II epithelial cells which in turn activate iNKT cells and secondarily other immune cells. Adenosine released from hypoxic tissues, or RA infused as an anti-inflammatory agent decrease proinflammatory cytokines and may be useful for treating cytokine storm in patients with Covid-19 or other inflammatory lung diseases or trauma.

## Introduction

Infection with Severe Acute Respiratory Syndrome Corona Virus 2 (SARS-CoV-2) caused Coronavirus disease 2019 (COVID-19). Most infected people exhibited a rapid IFN-1 response that controlled viral replication [1]. In some cases, persistence of the delta-variant that was common in the US at that time of this study in 2021 was associated with lung inflammation and injury [2]. Hospitalization was required in 1–3% of infected unvaccinated individuals. Hospitalized patients usually exhibited hypoxia and other systemic responses including procoagulant state, cytokine storm and epithelial barrier dysfunction [3]. About 20% of hospitalized patients progressed to acute respiratory distress syndrome (ARDS), which carried a 75% mortality rate [4]. Anti-inflammatory treatment strategies have been targeted to COVID-19 patients who exhibit lung inflammation. These treatments include steroids [5] and agents that selectively block IL-1 [6], IL-6 [7], or Janus-kinase (JAK) [8]. Adenosine, which binds to A1, A2A, A2B and A3 adenosine receptors, principally activates A2A receptor to produce broad anti-inflammatory effects. Adenosine reduced mortality when administered as an aerosol to intubated Covid-19 patients [9]. However, adenosine is rapidly metabolized in blood (half-life is < 1 second [10]) and it activates the A1 adenosine receptor to reduce heart rate and conduction velocity [11]. RA selectively activates the A2A adenosine receptor subtype, has a longer half-life than adenosine and does not cause heart block. RA marketed as Lexiscan™ or Rapidscan™ is a coronary vasodilator that is used for myocardial perfusion imaging. A2AR agonists such as RA also suppress proinflammatory cytokine release from most immune cells and consequently produce a broader array of anti-inflammatory effects than drugs that target a single cytokine [12].

A key target of RA is the iNKT cell that, unlike conventional T effector cells, is rapidly activated by tissue hypoxia or ischemia to produce and release pro-inflammatory cytokines. iNKT cells comprise a small CD1d-restricted subset of T cells that express semi-invariant TCRs that can be activated by certain bacterial glycolipids and/or by endogenous glycolipids derived from injured tissues. IL-33 released from SARS-CoV-2-infected epithelial cells facilitates iNKT cell activation via Sp2 receptors [13]. Depending on how they are activated, iNKT cells may secrete Th1, Th2 or Th17 cytokines such as IFN-γ, IL-14 and IL-17A, respectively [14, 15]. IFN-γ derived from iNKT cells perpetuates inflammation by stimulating production of IFN-γ-inducible CXCR3 chemokines, CXCL9, CXCL10 and CXCL11 [14, 16]. Expression of the activation marker CD69 on iNKT cells of COVID-19 patients at the time of hospital admission is predictive of clinical course and the severity of hypoxia [17].

Stressed cells release adenine nucleotides, primarily ATP, into the extracellular space [18, 19]. Extracellular ATP and ADP are metabolized to AMP and adenosine by ecto-enzymes including ectonucleoside triphosphate diphosphohydrolases (ENTPDs; including CD39),

and ecto-5'-nucleotidease (e5NT; CD73) [20]. Induction of these enzymes in response to hypoxia accelerates the conversion of pro-inflammatory ecto-ATP to anti-inflammatory ecto-adenosine [21, 22]. By binding to A2ARs, adenosine produces anti-inflammatory effects in iNKT cells as well as macrophages [23], dendritic cells [24], neutrophils [25] and platelets [26].

We hypothesized that A2AR activation would inhibit cytokine storm and reduce the pro-coagulation state in COVID-19 patients. Here we show in a pilot clinical safety study that infusion of RA into 5 hospitalized COVID-19 patients produced rapid subjective symptom improvement, increased SpO$_2$, reduced D dimer, produced a downward trend in CRP, reduced iNKT cell activation (% CD69+), and a reduced 11 of 13 pro-inflammatory cytokines measured in plasma. Additionally, RA administered in Alzet pumps for 7 days increased survival of K18-hACE2 mice infected with SARS-CoV-2. Further investigation of RA and other A2AR agonists as treatments for COVID-19 and possibly other pulmonary insults that cause SARS is warranted.

## Methods

This phase 1b study evaluated the safety, tolerability, and efficacy of RA for the treatment of hospitalized adults with moderate to severe COVID-19. The trial was conducted at the University of Maryland Medical Center, Baltimore, MD.

### Eligibility

**Inclusion criteria.**

1. 18 years of age or older

2. Laboratory-confirmed SARS-CoV-2+ by RT-PCR

3. Moderate to Severe COVID-19/WHO stage 4 or 5: Moderate illness is defined as individuals who have evidence of lower respiratory disease by clinical assessment or imaging and a saturation of oxygen (SpO2) >93% on room air at sea level. Severe Illness is defined as individuals who have respiratory frequency >30 breaths per minute, SpO2 $\leq$ 93% on room air at sea level, ratio of arterial partial pressure of oxygen to fraction of inspired oxygen (PaO2/FiO2) <300, or lung infiltrates >50%

4. Verbal or written informed consent from patient or legally authorized representative

**Exclusion criteria.**

1. Pregnant or breastfeeding

2. Signs or symptoms of acute myocardial ischemia or a cardiac intervention within the past 90 days.

3. Chronic cardiac conditions including non-vascularized coronary artery disease, heart failure, valvular disease, or cardiomyopathy.

4. Cardiac arrhythmia such as Sinoatrial (SA) or Atrioventricular (AV) Nodal Block/dysfunction, bradycardia, permanent pacemaker, internal defibrillator, or Atrial Fibrillation/Atrial Flutter requiring treatment or observation.

5. History of hypotension (sustained systolic blood pressure < 80 mmHg)

6. History of severe hypertension not adequately controlled with anti-hypertensive medications (Systolic blood pressure $\geq$ 200 mmHg and/or Diastolic blood pressure $\geq$ 110 mmHg)

7. Severe or moderate renal impairment or end stage renal disease (defined as GFR < 60 mL/min/1.73 m2)

8. History of clinically overt stroke within 3 years

9. History of seizure disorder

10. Pre-existing respiratory conditions, most notably asthma, chronic obstructive pulmonary disease or emphysema.

11. Respiratory failure for greater than 72 hours. Defined as the continuous use of mechanical ventilation, HFNC >20L/min, CPAP, and/or ECMO. (CPAP use due to obstructive sleep apnea is acceptable).

12. Treatment with chronic anti-coagulation or anti-platelet therapy (prophylactic aspirin is acceptable)

13. Treatment within 30 days of investigational agents as part of a research study.

14. Treatment with theophylline or aminophylline within 12 hours of study dosing

15. Treatment with Persantine and/or Aggrenox within 5 days

16. Clinical conditions that in the opinion of the investigator would make the subject unsuitable for the study.

## Recruitment

All COVID-19 clinical research studies at the UMB medical center are overseen by the institution's COVID-19 task force committee that organized the screening of COVID-19 patients for enrollment in clinical trials. This group functioned beginning in the Spring of 2020. A daily research "huddle" facilitated this process and identified assigning study subjects to the most appropriate trials. A representative from our study team attended this meeting each day and did not encounter any barriers related to competing trials. Study staff screened the medical records of COVID 19 positive patients and the COVID Huddle call to determine if they were eligible for the RA study. Hospitalized COVID-19 patients were approached according to the unit policy (video chat etc). Incentives were not offered to study participants. A legally authorized representative (LAR) was identified in case patients became incapacitated.

## Patient treatment

Infusion of RA at a rate of 1.44 μg/kg/h for 12–48 hours is safe in patients with sickle cell anemia experiencing painful vaso-occlusive episodes [27] and in lung-transplant recipients [28]. In order to establish that RA is safe in Covid-19 patients, we were initially directed by our Data and Safety Monitoring Board (DSMB) to treat 5 patients hospitalized Covid-19 patients with drug. The current study reports the effects of RA in this cohort, as well as survival data in mice with Covid-19. Having found that RA is safe, we have initiated a second double-blind randomized cohort, not included here to better assess drug efficacy. RA was administered intravenously by a trained registered nurse using a pediatric infusion pump with a loading dose of 5 μg/kg/h for 30 min followed by 1.44 /kg/h for 6h. Compared to Lexiscan that is rapidly injected at a rate of 400 μg/ 10 seconds, the slow infusions used in the current study were found to eliminate unpleasant side effects including headache, dizziness, nausea, stomach discomfort, mild chest discomfort, shortness of breath, and flushing. Oxygen saturation (SpO$_2$), blood pressure, heart rate, respiratory rate, CBC w/diff, CMP, sedimentation rate, creatine kinase, ABG, chest imaging, and EKG were monitored periodically throughout infusion.

## Blood collection and processing

Blood samples were drawn as a source of plasma and peripheral blood mononuclear cells (PBMCs). Blood samples were collected just prior to the start of RA infusion (0 hr), just prior to the end of the loading dose (30 min), 4 hours into the maintenance dose (4.5 hr), and the next day (≈ 24 hr after the start of infusion). After centrifugation at 366g for 10 minutes at room temperature (RT) plasma (1.5 ml) was collected and stored at -80°C for later analysis of cytokines, CRP, and D-dimer. PBMCs were isolated by the common Ficoll-Paque method with modifications. Plasma was replaced by 1.5 ml PBS and the blood was overlaid on 2.5ml Ficoll Paque Plus (GE Healthcare) in a sterile 15 ml conical tube and centrifuged at 693g for 25 minutes at RT without breaking. Cells at the interface were collected and transferred to 15mL conical tubes and washed twice with 12ml PBS and centrifuged at 366g for 7 minutes. Live cells were frozen overnight in freezing media (90%FBS+10%DMSO) at a concentration of $1X10^7$ cells/ml using a Mr. Frosty container at -80°C and transferred to liquid nitrogen the following morning.

## Flow cytometry

Live cells were thawed in 15 mL conical tubes prefilled with 10 ml warm (37°C) wash media (20% FBS-RPMI-1640) mixed, centrifuged at 500g for 10 minutes at RT, washed in PBS and centrifuged again. Cells were resuspended at $1X10^7$ cells/ml and transferred in 100 μl to 96 well plates for cell viability staining with Zombie NIR. Cells were then treated with human Fc block and stained with antibodies (Table 1) at 4°C for 20 minutes. Samples (including single positive controls, FMOs, and stained samples) were fixed with 2% paraformaldehyde and analyzed with a Cytek Aurora in the Flow Cytometry Core Laboratory (UMB, Baltimore, MD). Data were analyzed using FCS Express 7 Research Edition (De Novo Software, Pasadena, CA).

## Measurement of cytokines/chemokines

Plasma levels of cytokines/chemokines were measured as previously described [28] using a Luminex assay kit (Millipore Sigma, Burlington, MA) in the Cytokine Core Laboratory UMB.

## Treatment with RA in a murine COVID-19 model

Mouse studies were approved by the University of Virginia Animal Care and Use Committee (Protocol Number 4309) and performed in the University's certified animal Biosafety Level 3 laboratory. All surgery was performed under sodium pentobarbital anesthesia. Male 2-3-month-old 25–30 g B6.Cg-Tg(K18-ACE2) 2Prlmn/J mice were obtained from Jackson Laboratories [29]. These mice express the SARS-CoV-2 receptor, human angiotensin-converting enzyme 2 (hACE2) in epithelia and develop a usually lethal infection after intranasal inoculation with a human strain of virus [30, 31]. To treat COVID-19 in mice treated with RA, 7-day

**Table 1. Antibodies used for flow cytometry experiments.**

| Specificity | Clone | Fluorochrome | Target | Vendor |
|---|---|---|---|---|
| CD3 | UCHT1 | BV 605 | Lineage T Cells | Biolegend |
| CD19 | SJ25-C1 | PE-Ax610 | B Cells | Fisherthermo |
| CD69 | FN50 | Ax 700 | Activation | Biolegend |
| TCR Vα24-Jα18 | 6B11 | PerCP-eF710 | INKT Cells | Invitrogen |
| Zombie | NA | NIR | Viability | Biolegend |

Ax, Alexa; NIR, NIR dye for dead cell stain; BV, brilliant violet; PE, R-phycoerythrin; PerCP, peridinin chlorophyll protein.

Alzet 1007D osmotic pumps with a pump rate of 0.5 μl/hr and containing saline or 80 μg/ml RA (≈ 1.44 μg/kg/h) were implanted subcutaneously just prior to being challenged with 1250 Pfu of Hong Kong VM20001061/2020 (NR-52282, BEI Resources) via the intranasal route under ketamine/xylazine anesthesia. Mice were evaluated twice daily for clinical symptoms, including weight loss, activity level, fur appearance, posture, and eye closure and were euthanized when weight loss was >20% of initial weight or a maximum score was observed in two symptom categories.

## Statistical analysis

We normalized human study data (NKT cell activation, cytokines, etc) to baseline levels sampled before the start of drug treatment. Changes in patient circulating cytokine levels over time were analyzed using a repeated measures model with a spatial power covariance structure. Data were analyzed using a log scale to facilitate interpretation as fold changes relative to baseline. Ninety-five % confidence intervals were used to estimate changes from baseline at 30 minutes, 4.5 hours and 24 hours. Blood pressures were analyzed using repeated measures applied to changes from baseline. The repeated measures analyses were carried out in SAS 9.4 PROC MIXED. Clinical data (SpO$_2$, CRP and D-dimer) were reported as mean ± SEM; paired t-tests were used to compare the 24 hour levels to baseline. GraphPad Prism software version 9 was used for statistical analysis of flow cytometry data; paired t-tests were used to compare percentages of circulating CD69+CD3+6B11- conventional T cells and CD69+6B11+ iNKT cells at baseline and 30 minutes into RA infusion. Results were considered statistically significant if P<0.05. Differences between mouse survival curves are based on the Log rank Mantel Cox test.

## Ethics statement

Human subjects: This study was approved by the institutional review board of the University of Maryland, Baltimore (IRB number: HP-00091372) and registered at Clinicaltrials.gov (NCT04606069). RA treatment of moderate to severe COVID-19 patients was approved by FDA (IND # 149635).

## Results

Case Series: Subjects participating in this study were patients requiring hospital admission for COVID-19 diagnosed by PCR and provided informed written consent to receive intravenous RA. Based on prior studies, the steady-state plasma level of RA after infusion for 6 hours of 1.44 μg/kg/h is about 2 ng/ml and is devoid of side effects [31]. By comparison, RA administered as Lexiscan, a 400 μg bolus over 10 seconds for myocardial perfusion imaging results in a peak plasma level of about 15 ng/ml, and is associated with unpleasant side effects [32, 33].

The 5 subjects enrolled in this study were all unvaccinated and classified as WHO stages 4–5 [34]. Based on their clinical need they were given supplemental oxygen by nasal canula (NC) including high flow NC, but none required intubation. All 5 patients recovered and were discharged on room air 3–9 days after admission. Upon diagnosis of COVID-19, all patients received 6 mg daily dexamethasone as standard of care. RA was administered within 3 hours of consent. Side effects were minimal; there was 1 case of transient self-limiting nausea, but no other toxicities or reportable side effects during infusion and 30 days of follow-up. None of the subjects who consented to the study dropped out. Fig 1 summarizes subject enrollment, allocation, follow-up, and analysis. Table 2 summarizes patient demographics. Although RA at high doses can produce vasodilation mediated by A2ARs on vascular smooth muscle, at the dose used in this study blood pressure was minimally changed as illustrated in Fig 2.

## COVID-19 CONSORT Flow Diagram

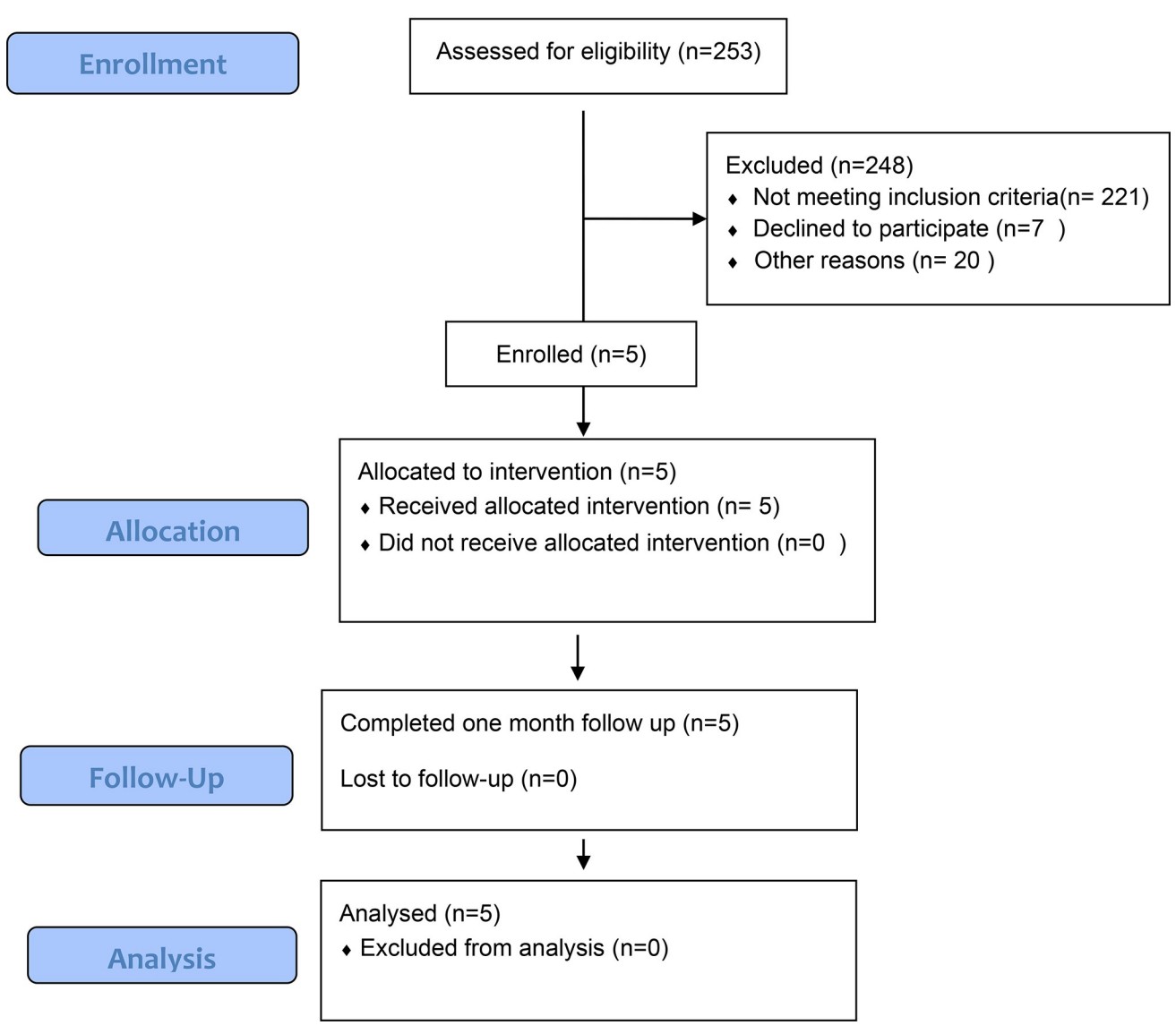

**Fig 1. Subject flow diagram.**

**Table 2. Demographics of the COVID-19 patients who received regadenoson treatment.**

| Subject ID | Age | Gender | Race | Ethnicity | BMI | Oxygen Supplementation on Admission |
|---|---|---|---|---|---|---|
| 101–001 | 60 | M | AA | Not Hispanic or Latino | 43.7 | 2L NC |
| 101–002 | 46 | F | NA | Not Hispanic or Latino | 36.3 | 2L NC |
| 101–003 | 37 | M | AA | Not Hispanic or Latino | 40.2 | 2L NC |
| 101–004 | 65 | M | W | Not Hispanic or Latino | 29.2 | 40L HFNC FiO2 60% |
| 101–005 | 58 | M | W | Not Hispanic or Latino | 28.2 | 6L NC |

M: Male; F: Female; AA: African American; NA: Native American; BMI: body mass index; L: Liters; NC: nasal cannula; HFNC: High flow nasal cannula

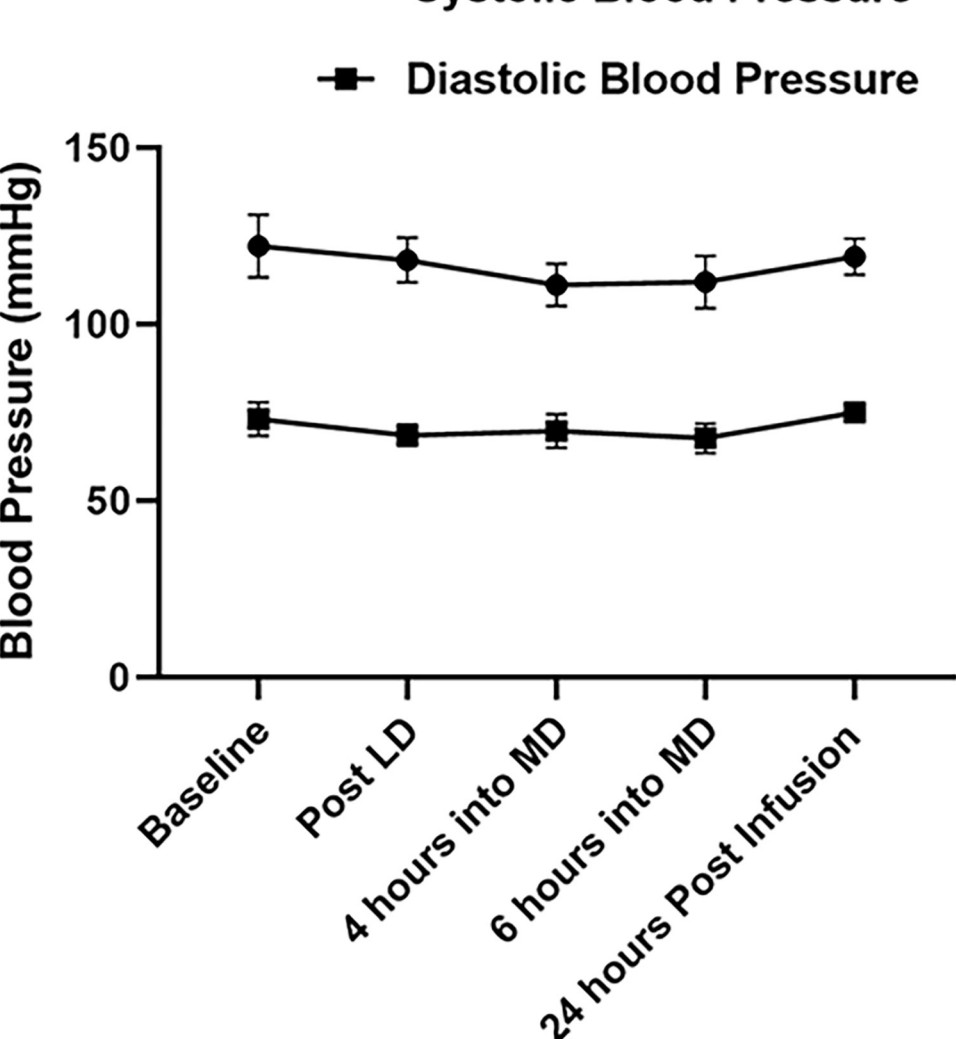

**Fig 2. Changes in blood pressure before, during and after regadenoson (RA) infusion.** LD: Loading dose, 2.5 µg/kg over 30 minutes. MD: Maintenance dose,1.44µg/kg/hour for 6 hours. Data shown are means ± SEM from 5 RA treated COVID-19 patients. Repeated measures analysis applied to changes from baseline shows no significant changes during or after RA infusion (SBP, p = 0.28, DBP, p = 0.28).

**Case 1:** A 60-year-old obese male [body mass index (BMI), 43.7] with a history of hypertension and gastric reflux presented with 4-days of chest pain and shortness of breath. On admission, oxygen saturation ($SpO_2$) was 90% and chest x-ray demonstrated patchy opacities/consolidation at the bases. The patient received standard of care treatments including Remdesivir and dexamethasone with 2 liters/min supplemental $O_2$ (WHO stage 4). He was treated with RA one day after admission and reported rapid subjective symptomatic improvement. $O_2$ support was reduced to 1 liter NC 2 days later and the patient was discharged on day 6.

**Case 2:** A 46-year-old obese female (BMI 36.3) with a history of hiatal hernia and diverticulitis, presented with a 10-day history of symptoms including shortness of breath with cough, and later with nausea and vomiting. Prior to admission the patient received steroids and non-

standard treatment with Albuterol, Azithromycin, and Ivermectin. On admission, the patient was febrile (39.4°C) and hypoxic (SpO$_2$ 85%). She received standard treatments including dexamethasone and 2 liters/min of O$_2$ by NC (WHO stage 4). The patient was treated with RA one day after admission and reported rapid subjective symptomatic improvement. O$_2$ support was reduced to 1 liter within 2 days and the patient was discharged on day 6.

**Case 3:** A 37-year-old obese male (BMI 40.2) with no other medical history presented after 7 days of shortness of breath on exertion that limited his physical activity. On admission the patient was hypoxic with a SpO$_2$ of 88% and his chest x-ray demonstrated right lower lobe opacities. He received standard treatments including Remdesivir and dexamethasone with 2 liters/min O$_2$ by NC. The patient was treated with RA on the day of admission. After the RA infusion, his SpO2 increased to 97%, and his oxygen supplementation was weaned off with symptomatic improvement the following day. The patient was discharged on day 3.

**Case 4:** A 65-year-old male (BMI 29.2) with a remote smoking history presented with persistent symptoms of shortness of breath after outpatient treatment with monoclonal antibodies. On admission the patient was hypoxic with a SpO$_2$ of 87%. His chest x-ray showed bilateral patchy opacities with mild to moderate pulmonary edema while a chest CT demonstrated peripheral ground glass opacities with destructive changes of the dependent parenchyma, consistent with COVID-19 pneumonia. He received treatment including a JAK inhibitor and dexamethasone while also requiring 40L/minute high flow HFNC with a FiO$_2$ of 60% (WHO stage 5) and was admitted to the medical intensive care unit (MICU). The patient was treated with RA after admission with rapidly improved SpO$_2$ by completion of the RA infusion. The patient was transferred out of MICU on day 3, weaned down to nasal cannula support by day 7 and discharged on day 9.

**Case 5:** A 58-year-old male (BMI 28.19) with a history of gastric reflux presented with persistent cough, shortness of breath, and acute hypoxia. His SpO$_2$ was 88% on admission and he was placed on 6 liters/minute of O$_2$ via nasal cannula (WHO stage 4). His chest x-ray was positive for multifocal patchy opacities with low lung volumes, while a chest CT demonstrated widespread peripheral ground glass opacities, consistent with COVID-19 pneumonia. He was out of the window for remdesivir but received dexamethasone and continued oxygen supplementation. The patient received RA infusion and his SpO$_2$ improved during treatment. He reported a self-limiting episode of nausea that resolved after treatment was complete. The patient was discharged on day 9.

## Evidence that RA reduces lung inflammation

Oxygen saturation (SpO$_2$) by pulse oximetry was measured in all 5 subjects before admission (without oxygen supplementation), after admission (after starting oxygen supplementation), and periodically after RA infusion. Compared to SpO$_2$ measured prior to the start of RA infusion (baseline), SpO$_2$ was significantly elevated at 24 hours post RA infusion (93.80 ± 0.58 vs 96.60 ± 1.08, P = 0.031) (Fig 3A). CRP levels were high (>3 mg/L) in 3 of 4 patients (measured only in 4 of 5 patients) and were reduced by >50% 24 hours after RA infusion (3.80 ± 1.40 vs 1.98 ± 0.74 mg/dL, P = 0.075) (Fig 3B). D dimer levels were high (>500 ng/ml) in 3 of 5 patients and were decreased at 24 hours post RA infusion (754 ± 168 vs 518 ± 98.3, P = 0.043) (Fig 3C). Our analysis did not account for the possibility that some of these changes may have occurred spontaneously over 24 h.

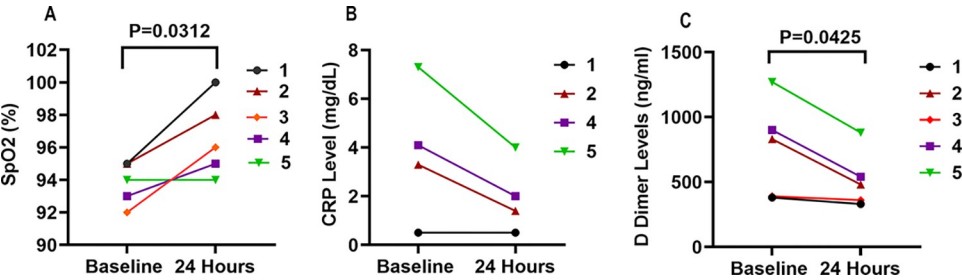

**Fig 3. Effects of regadenoson (RA) infusion on oxygen saturation and C-reactive protein (CRP) levels in plasma of COVID-19 patients.** (**A**) SpO2 was determined by pulse oximetry just prior to and 24 hours after the initiation of 6.5 hours of RA infusion. There was a statistically significant difference in in SpO2 based on paired T-test (p = 0.03,n = 5). (**B**) CRP before and 24 h after RA infusion, P = 0.075 (n = 4). (**C**) D-dimer before and after RA infusion (p = 0.043, n = 5).

## Invariant NKT cells are influenced by COVID-19 and RA

Cytokine storm syndrome is characterized by elevated levels of proinflammatory cytokines, including IFN-$\gamma$, TNF$\alpha$, and IL-17 [35, 36]. Alarmins such as IL-33 are tissue-derived nuclear proteins constitutively expressed at high levels in epithelial and endothelial barrier tissues. IL-33 released from damaged epithelial cells may be important for initiating the inflammatory response that can lead to cytokine storm since it is elevated in COVID-19 patients and can activate iNKT cells [37]. As shown in Fig 4A, we identify rare iNKT cells (comprising 1% of all CD3+ T cells in blood) by flow cytometry and we identified an activated subset based on expression of the early lymphocyte activation marker, CD69. We found that iNKT cells, but not conventional T cells from COVID-19 patients are mostly CD69+ (Fig 4B). Moreover, the data show for the first time that RA treatment rapidly reduces the fraction of circulating iNKT cells that are CD69+ by 50% within 30 minutes (Fig 4C). These findings suggest that cytokine storm in COVID-19 is initiated in part by IL-33, and possibly other alarmins released from infected epithelial cells, and trigger iNKT cell activation due to IL-33 binding to its SP2 receptor.

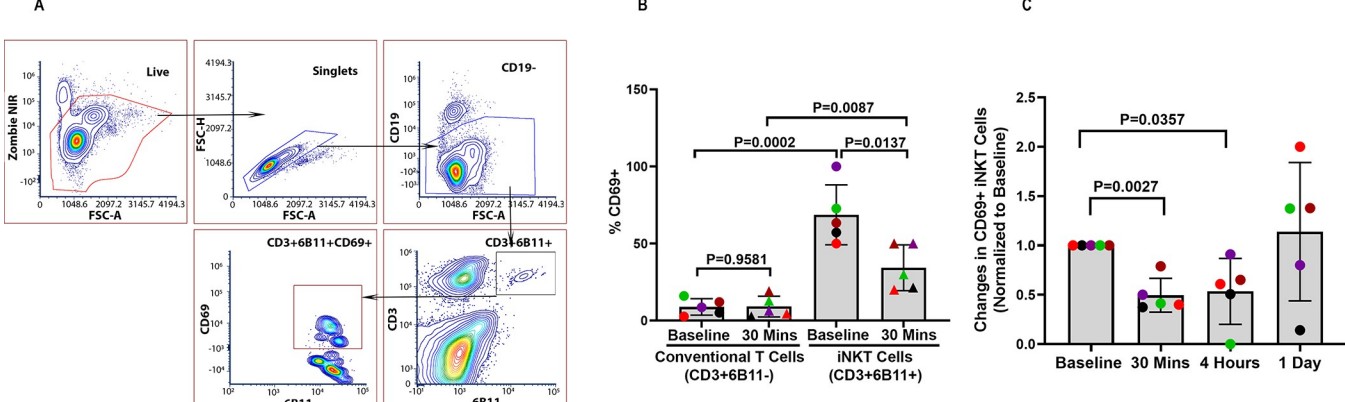

**Fig 4. Decrease in circulating iNKT cell activation during regadenoson (RA) infusion into patients with COVID-19.** (**A**) Flow cytometry gating strategy used to identify CD3+6B11+iNKT cells and their activation-state based on CD69 expression. (**B**) Comparison of percentages of circulating CD69+CD3+6B11- conventional T cells and CD69+6B11+ iNKT cells at baseline and 30 minutes into RA infusion. (**C**) Changes in the activation state of iNKT cells in blood of COVID 19 patients before, during and after RA infusion. PBMCs prepared from blood samples of 5 subjects were thawed and analyzed by flow cytometry at the same time. The color scheme used in Fig 4B & 4C denote the same individuals as shown in Fig 3A.

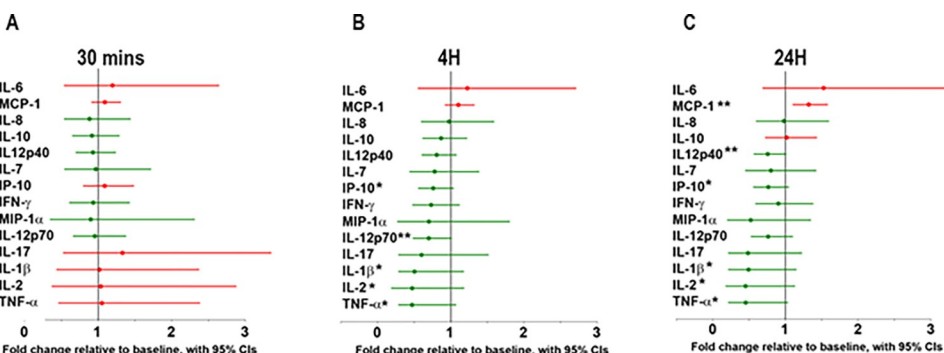

**Fig 5. Changes of plasma proinflammatory cytokines/chemokines during and after RA infusion.** Cytokine levels were normalized to baseline just prior to RA infusion. Cytokine levels below and above baseline values are depicted by *green* and *red*, respectively. Blood was collected before RA infusion (baseline) and 3 times after the start of infusion: (**A**) at the end of loading dose (5μg/kg/hour, 30 minutes), (**B**) 4 hours into the maintenance dose (1.44μg/kg/hour, 4.5 hours) and (**C**) 17 hours post RA infusion, n = 5. Green dots and lines indicate cytokine levels lower than baseline, *$P < 0.1$; **$P \leq 0.05$ relative to baseline by Paired T-Tests.

A reduction in the activation of iNKT cells in response to RA infusion is expected to reduce the production of proinflammatory cytokines produced by iNKT cells and other cells. We compared the ratio over baseline of plasma cytokines at 30 minutes, 4.5 hours, and 24 hours after the start of RA infusion (Fig 5). RA had little effect on the levels of 13 pro-inflammatory cytokines at 30 minutes (Fig 5A), but 11 of 13 were reduced at 4.5 hours (Fig 5B) and 24 hours, (17 h hours after the end of RA infusion, Fig 5C). The data suggest that suppression of iNKT activation by RA precedes changes in plasma cytokines.

## Treatment with regadenoson increases survival of SARS-CoV-2-infected mice

The current study was not designed to determine if RA reduces mortality or length of stay in hospitalized COVID-19 patients. In order to investigate the effects of RA on survival we evaluated K18-hACE2 mice [38]. Thirty mice infected with SARS-CoV-2 were treated by subcutaneous infusion with RA and 11 controls were infused with saline. As shown in Fig 6, over 90% of control mice died by day 10, mostly on days 4–5 after infection. Only 50% of RA treated mice died, mostly on day 7, at the time when RA ran out. These findings show that RA infusion decreases mortality in COVID-19-infected mice and suggest that longer treatment may produce better protection.

## Discussion

For this case series we treated five COVID-19 patients with an intravenous infusion of RA. These patients were enrolled in an NHLBI sponsored Clinical Trial (NCT0460606) to evaluate safety and evidence of efficacy of RA for reducing cytokine storm. This is the first examination of an A2AR selective agonist in COVID-19 patients. We restricted this initial study to moderately ill patients as characterized by respiratory disease based on clinical assessment and radiographic imaging in addition to oxygen supplementation with either low flow (WHO stage 4) or high flow (WHO stage 5). No intubated or ECMO patients (WHO stages 6–7) were enrolled.

Adenosine is produced by hypoxic or ischemic tissues and has been found to have vasodilatory, anti-platelet and broadly anti-inflammatory effects that are mostly mediated by the A2AR. RA is a selective A2AR agonist that is clinically used as a coronary vasodilator for

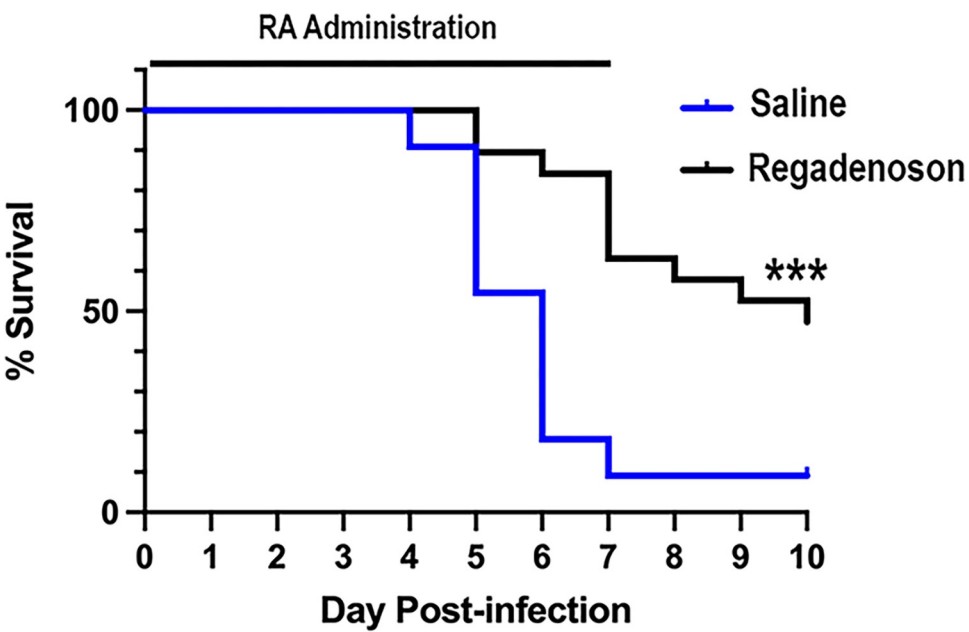

**Fig 6. Regadenoson increases survival of SARS-CoV-2-infected K18-mice.** Seven day 0.5 μl/h Alzet pumps loaded with 80 μg/ml Regadenoson (N = 19), or saline (N = 11) were implanted subcutaneously into 2–3 month old B6.Cg-Tg (K18-ACE2)2Prlmn/J mice just prior to infection with 1250 Pfu of Hong Kong VM20001061/2020. Mice were euthanized based on criteria described in Methods. Data are combined from three independent experiments. Survival curves are statistically different based on the Log rank Mantel Cox test, P<0.001. The time of RA infusion is indicated by a horizontal bar.

myocardial perfusion imaging. During myocardial perfusion imaging RA frequently produces unpleasant side effects. Adverse events begin soon after dosing and quickly resolve within an hour in most patients. In the current study we avoided the side effects associated with 400μg bolus injection by administering a slow loading dose of 5 μg/kg/h over 30 minutes followed by a maintenance does of 1.44 μg/kg/h for 6 h. We have previously reported that RA is safe and devoid of side effects during 12 and 48 hours of continuous infusions of 1.44 μg/kg/h in sickle cell patients undergoing painful vaso-occlusive crises [39] and lung transplant recipients [28].

Of the 5 COVID-19 patients treated with RA in this study, only one experienced a mild side effect, transient self-limiting nausea. We conclude that RA is safe to use in moderately ill COVID-19 patients. We show that similar dosing of RA to K18-hACE2 transgenic mice that are susceptible to COVID-19 causes a significant reduction in mortality.

This uncontrolled 5 patient case series was not powered to demonstrate that RA reduces mortality or length of stay in the hospital. We did observe promising effects within 24 hours of RA treatment including: subjective improvement in all subjects; increased SpO2; decreased D-dimer selectively in patients with high D-dimer levels (>500 ng/ml); a trend toward lower CRP selectively in patients with high CRP levels (> 3mg/L); rapidly reduced expression of CD69 on circulating iNKT cells; and a gradual reduction in circulating levels of 11 of 13 pro-inflammatory cytokines. A larger placebo-controlled clinical trial is needed to confirm beneficial effects of RA. The current study does suggest that RA will be effective for rapidly reducing iNKT cell activation and cytokine production in COVID-19 patients.

In previous animal studies the administration of A2AR agonists was found to reduce ischemia reperfusion-injury (IRI) in several tissues including liver [40] and lung [28, 41]. Although A2AR activation produces anti-inflammatory responses in neutrophils, macrophages, T cells and platelets, the cell that appears to be most important for adenosine to suppress

inflammatory cascades is the iNKT cell [42, 43]. The activation of iNKT cells is probably mediated in part by epithelial cell damage from SARS-CoV-2 infection to produce host lipid mediators that activate the invariant TCR, and/or by the release of IL-33 that is elevated in COVID-19 patients and activates iNKT cells via SP2 receptors [13]. Fig 7 illustrates some of the processes that may be involved in activating iNKT cells in COVID-19 patients, and their inhibition by RA.

The activation of A2ARs on platelets inhibits their adhesion and aggregation [26, 44]. Blood levels of D-dimer have been shown to be highly correlated with unfavorable COVID-19 outcomes [45], development of pulmonary emboli [46, 47] and vascular thrombotic complications [48]. RA infusion is associated with reduced D-dimer levels. This finding suggests that circulating RA may reduce the risk of pulmonary embolism and vascular thrombotic complications. Further study is necessary to prove this hypothesis.

Proinflammatory cytokine transcription is stimulated in iNKT cells in response to NF-κB activation [42]. A2AR transcription is slowly enhanced in response to NF-κB activation and RA evokes an anti-inflammatory response [42]. Over time after tissue injury, A2AR expression and signaling become more effective at suppressing the activation of iNKT cells and other leukocytes. The early activation marker CD69 is expressed at low levels in health individuals, but is elevated on iNKT cells of COVID-19 patients. The fraction of CD69+ iNKT cells at the time of admission correlates with patient hypoxia and poor outcome [17]. Other treatments that might be expected to reduce the inflammatory response in COVID-19 patients are anti-iNKT or anti-CD1d antibodies [49, 50] to deplete iNKT cells or block their activation, respectively.

As previously noted by Correale et al [9] aerosolized adenosine appears be effective for treating COVID-19 subjects receiving supplemental oxygen. Iatrogenic hyperoxia therapy reduces endogenous extracellular adenosine production. Moreover hyperoxia may reduce the induction of A2ARs in iNKT cells [42], T cells [51], and macrophages [23]. Other factors that may reduce adenosine production and signaling in the hyperoxic lung are reduced expression of the ecto-enzymes CD39 and CD73 [21, 22] and inactivation of PANX1 channels that release ATP [18, 52]. These observations suggest that hyperoxia will suppress adenosine production and signaling due to several factors that may be counteracted by RA infusion.

In the current study RA was administered by IV infusion and gained access to blood and peripheral tissues. In the study by Correale et al. [9] hospitalized COVID-19 patients inhaled adenosine once or twice daily for 5 days. Despite its short half-life adenosine-treated patients exhibited a significant increase in $PaO_2/FiO_2$ and death rate was reduced by adenosine inhalation. Beneficial effects of inhaled adenosine were also reported in a subsequent two patient case study [53]. Inhaled adenosine appears to be metabolized rapidly and does not accumulate in peripheral blood since cardiovascular effects noted with IV adenosine are absent [54]. Unlike RA, adenosine activates A1 adenosine receptors in the cardiac SA and AV nodes, which can produce bradycardia or heart block. Compared to inhaled adenosine, which has a short half-life and does not accumulate in blood, RA has a terminal half-life of 2 hours, is distributed throughout the body and likely produces generalized anti-inflammatory and anti-coagulant effects.

## Limitations of the study

Since the effects of RA in COVID-19 patients were unknown, we applied strict exclusion criteria to ensure patient safety. We assessed 253 hospitalized COVID-19 patients, 221 did not meet inclusion criteria. We enrolled 5 COVID-19 patients to receive RA. This study did not include untreated patient controls or have sufficient power to determine if RA reduced mortality or length of stay in the hospital, but we did observe promising effects in all five subjects

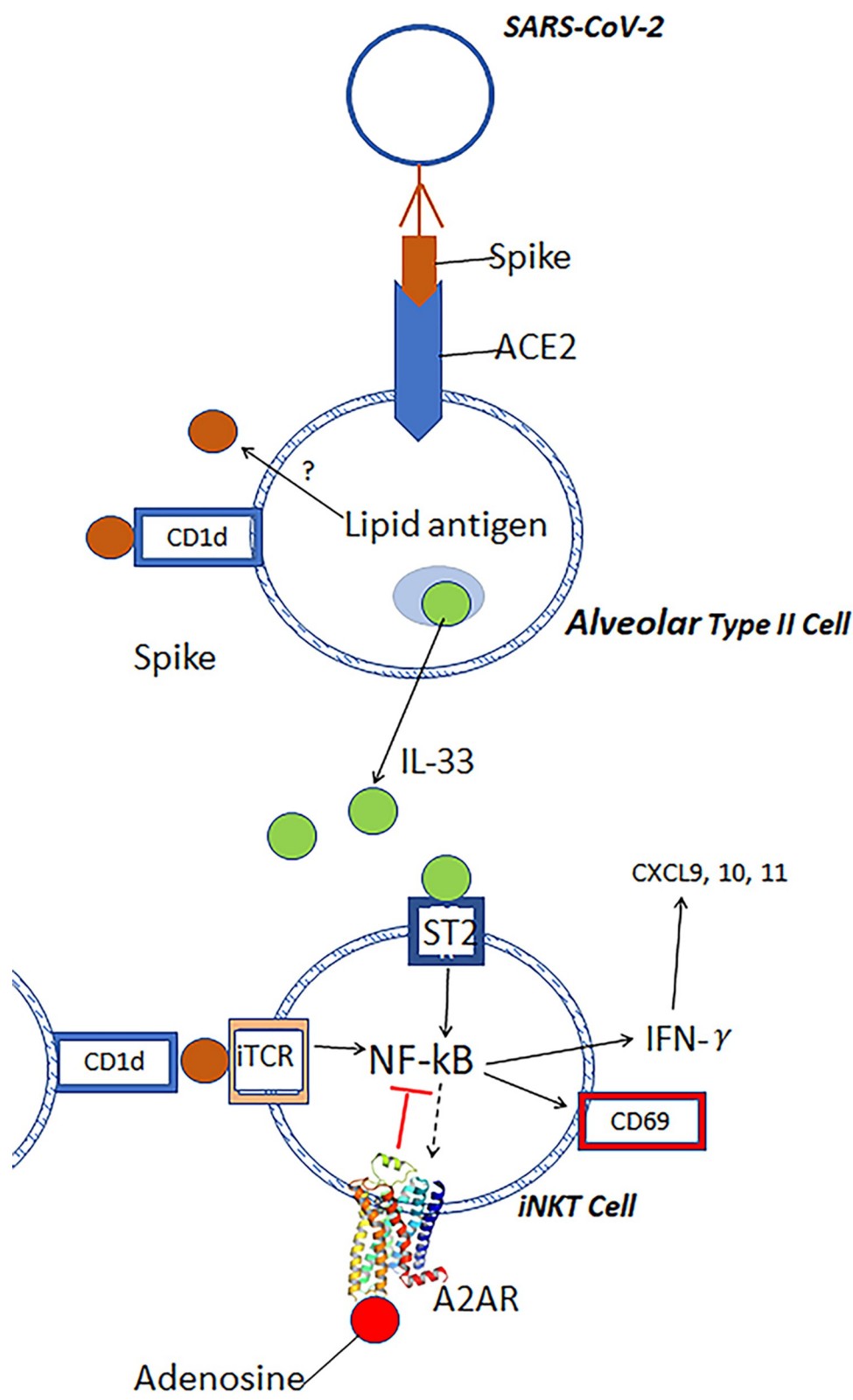

**Fig 7. Diagram showing some of the factors that mediate cytokin storm following infection with COVID-19.** The SARS-CoV-2 virus gains entry into lung type II epithelial cells by binding its spike protein to Angiotensis Converting Enzyme 2. CD1d is expressed by epithelial cells. Viral entry causes stressed cells to release alarmins such as IL-33 which

is elevated in the blood of COVID-19 patients. IL-33 and possibly CD1d-restricted lipid antigens activate NF-κB in iNKT cells and induce the rapid transcription of cytokines such as IFN-γ, IL-4 and IL-17. The release of IFN-γ propagates cytokine storm by stimulating the release from stromal and other cells of T cell attractants, CXCL9, CLCL10 and CXCL11. The activation of NF-κB in iNKT cells also causes a slow induction over 24–48 hours of A2ARs that enhance over time the anti-inflammatory potency of adenosine.

within 24 hours of RA treatment: subjective improvement; increased SpO2; a trend toward lower CRP levels; rapidly reduced expression of CD69 on circulating iNKT cells; and a gradual reduction in circulating levels of 11 of 13 pro-inflammatory cytokines. The results are not adjusted for multiple statistical tests and some of the observed reductions in cytokine levels might occur spontaneously over time. A larger placebo-controlled clinical trial is needed to confirm these findings, but they do suggest that RA will be effective for rapidly reducing iNKT cell activation and cytokine production in inflamed COVID-19 patients. The ability of RA to improve the survival of SARS-CoV-2-infected mice enhances out confidence in the results of the human studies.

## Conclusion

Oxygen supplementation in COVID-19 patients with poor lung function may cause an iatrogenic reduction in the release of adenine nucleotides into the extracellular space and a reduction in the rate of ATP conversion to adenosine by ecto-enzymes. These considerations provide a rationale for administering RA or other A2AR agonists during oxygen supplementation that suppresses ATP release from cells and metabolism to adenosine. In the current study we confirmed that iNKT cells are activated in COVID-19 patients and we show for the first time that their activation is rapidly reduced in response to RA infusion. The RA-induced reduction in iNKT cell activation precedes a decline in circulating levels of proinflammatory cytokines. Mouse studies demonstrate improved survival in COVID-19 infected animals treated with RA. These findings justify further evaluation of RA and other A2AR agonists for treatment of COVID-19 patients and other causes of lung inflammation.

## Supporting information

**S1 Checklist. TREND statement checklist.**
(PDF)

**S1 Table. Data of blood pressure from the 5 enrolled patients prior, during and post regadenoson infusion.**
(DOCX)

**S2 Table. Data of oxygen saturation (SpO2), D-Dimer and CRP from the participant patients prior and post regadenoson infusion.**
(DOCX)

**S3 Table. Changes of CD69+ T cells, CD69+iNKT Cells in the participated patients prior, during and post regadenoson infusion.**
(DOCX)

**S4 Table. Plasma levels of cytokine/chemokines (pg/ml) in the participant patients prior, during and post regadenoson infusion.**
(DOCX)

**S1 File.**
(PDF)

## Acknowledgments

We are grateful to Dr. Xiaoxuan Fan in the Flow Cytometry Core Laboratory (University of Maryland, Baltimore) for assistance with Flow Cytometry and Dr. Lisa Hester in the Cytokine Core Laboratory at UMB.

## Author Contributions

**Conceptualization:** Yunge Zhao, Joel Linden, Christine L. Lau.

**Data curation:** Yunge Zhao, Ezzat Mostafa, Emily Fleischmann, Kenneth L. Brayman, Alexander Krupnick.

**Formal analysis:** Mark R. Conaway, Preeti Chhabra.

**Funding acquisition:** Joel Linden, Christine L. Lau.

**Investigation:** Joseph Rabin, Yunge Zhao, Barbara J. Mann, Alexander Krupnick.

**Methodology:** Yunge Zhao, Preeti Chhabra, Kenneth L. Brayman.

**Project administration:** Manal Al-Suqi.

**Software:** Mark R. Conaway.

**Supervision:** Joseph Rabin, Joel Linden, Christine L. Lau.

**Validation:** Barbara J. Mann.

**Visualization:** Yunge Zhao.

**Writing – original draft:** Yunge Zhao, Joel Linden.

**Writing – review & editing:** Joseph Rabin, Yunge Zhao, Ezzat Mostafa, Joel Linden, Christine L. Lau.

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
