## [Decision Letter · Decision Letter 0]

14 Nov 2022

PONE-D-22-17321Regadenoson for the treatment of COVID-19: a five case clinical series and mouse studiesPLOS ONE

Dear Dr. Lau,

Thank you for submitting your manuscript to PLOS ONE. After careful consideration, we feel that it has merit but does not fully meet PLOS ONE’s publication criteria as it currently stands. Therefore, we invite you to submit a revised version of the manuscript that addresses the points raised during the review process. Your manuscript has been assessed by two peer-reviewers and their reports are appended below.  The reviewers comment that the manuscript could be strengthened with additional detail and clarifications. One of the reviewers comments that the manuscript may present incorrect information regarding the delivery of adenosine by V infusion. The statistical referee has commented on the statistical analysis reported, and indicates that the statistical limitations of the study should be discussed further in the discussions section.  Could you please carefully revise the manuscript to address all comments raised?

We look forward to receiving your revised manuscript.

Kind regards,

Maria Elisabeth Johanna Zalm, Ph.D

Editorial Office

PLOS ONE

Journal Requirements:

"We are grateful to Dr. Xiaoxuan Fan in the Flow Cytometry Core Laboratory (University of Maryland, Baltimore) for assistance with Flow Cytometry and Dr. Lisa Hester in the Cytokine Core Laboratory at UMB. This study was supported by NIH R01HL128492-05S1, the Manning Fund for COVID-19 Research,  MC10, and Astellas Pharma US, Inc.,  ISR005923. Patent pending for this manuscript."

"This study was supported by NIH R01HL128492-05S1 (CLL), the Manning Fund for COVID-19 Research, MC10 (JL), and Astellas Pharma US, Inc., ISR005923 (CLL & JL). The sponsors or funders play NO role in the study design, data collection and analysis, decision to publish, or preparation of the manuscript"

Reviewers' comments:

Reviewer's Responses to Questions

**Comments to the Author**

1. Is the manuscript technically sound, and do the data support the conclusions?

Reviewer #1: Yes

Reviewer #2: Yes

2. Has the statistical analysis been performed appropriately and rigorously? 

Reviewer #1: Yes

Reviewer #2: Yes

3. Have the authors made all data underlying the findings in their manuscript fully available?

Reviewer #1: No

Reviewer #2: Yes

4. Is the manuscript presented in an intelligible fashion and written in standard English?

Reviewer #1: Yes

Reviewer #2: Yes

5. Review Comments to the Author

Reviewer #1: Well written manuscript, however I feel the focus is much on outcomes of more interest in the larger trial as opposed to the safety focus that this manuscript should focus on.

Here are some comments:

Why are the primary and secondary endpoints of the randomized trial included as primary and secondary in this study of the five non-randomized patients treated with focus on the safety, tolerability and toxicity?

Regarding the statistical analyses of the continuous outcomes, they should in the larger trial account for the baseline value i.e. include the baseline value as a covariate. Change scores, normalization etc. does not appropriately adjust for baseline values as including it as a covariate does (much has been written about this in the literature). In merely 5 individuals there are not that many degrees of freedom to play with hence the results can be left as is. However, many of the results can simply be regression towards the mean e.g. if you have a high baseline value you will most likely have a lower value at the next measure and this should somehow be mentioned under limitations.

Please write under limitations that there is a risk of spurious results due to the number of statistical tests (it could somehow be combined with my previous comment)

The two way anova under figure 2 is not mentioned under methods

Reviewer #2: Overall this is an interesting paper describing the beneficial effects of a single infusion of Regadenoson, an A2A adenosine receptor agonist, on patients with COVID and murine data which further confirms the beneficial effects of this infusion. Moreover, the authors report that the principal antiinflammatory effects of regadenoson in this setting is mediated via inhibition of NKT cells. Overall, the clinical data remain quite preliminary and little can be said regarding efficacy of treatment in patients although the murine data is suggestive and prior clinical data using adenosine, a non-selective agonist, provides some evidence of efficacy in patients with COVID. There is one issue that should be corrected. The authors repeatedly note that adenosine cannot be given by IV infusion because of its effects on the heart (via A1 receptor stimulation). Adenosine is currently on the market for use as an infusion agent for pharmacologic stress testing so all of the references to A1 inhibitors and the contraindications to use of IV adenosine, etc should be eliminated.

6. PLOS authors have the option to publish the peer review history of their article (what does this mean?). If published, this will include your full peer review and any attached files.

Reviewer #1: No

Reviewer #2: **Yes: **Bruce N. Cronstein, MD

---

## [Author Response · Author response to Decision Letter 0]

10 Jan 2023

Editor Requirements:

Response: We have included all three required items and named them as “ Response to Reviewers”, “Revised Manuscript with Track Changes”, and “Manuscript”.

Journal Requirements:

Response: The revised files have been named to conform with journal requires. We have revised formatting of the main text , title and author affiliations. 

"We are grateful to Dr. Xiaoxuan Fan in the Flow Cytometry Core Laboratory (University of Maryland, Baltimore) for assistance with Flow Cytometry and Dr. Lisa Hester in the Cytokine Core Laboratory at UMB. This study was supported by NIH R01HL128492-05S1, the Manning Fund for COVID-19 Research, MC10, and Astellas Pharma US, Inc., ISR005923. Patent pending for this manuscript."

"This study was supported by NIH R01HL128492-05S1 (CLL), the Manning Fund for COVID-19 Research, MC10 (JL), and Astellas Pharma US, Inc., ISR005923 (CLL & JL). The sponsors or funders play NO role in the study design, data collection and analysis, decision to publish, or preparation of the manuscript"

Response: We have removed the funding information from the Acknowledgments and corrected the funding statement. 

Reviewer #1: Well written manuscript, however I feel the focus is much on outcomes of more interest in the larger trial as opposed to the safety focus that this manuscript should focus on.

Here are some comments:

1. Why are the primary and secondary endpoints of the randomized trial included as primary and secondary in this study of the five non-randomized patients treated with focus on the safety, tolerability and toxicity?

Response: Thank you for pointing out that the original manuscript lacked clarity on this point. We state in the revision that only five non-randomized patients are included in this report. In the revised manuscript we note that this five case report and mouse study provide evidence of Regadenoson safety and anti-inflammatory efficacy, although a definitive trial has yet to be completed.

2. Regarding the statistical analyses of the continuous outcomes, they should in the larger trial account for the baseline value i.e. include the baseline value as a covariate. Change scores, normalization etc. does not appropriately adjust for baseline values as including it as a covariate does (much has been written about this in the literature). In merely 5 individuals there are not that many degrees of freedom to play with hence the results can be left as is. However, many of the results can simply be regression towards the mean e.g. if you have a high baseline value you will most likely have a lower value at the next measure and this should somehow be mentioned under limitations.

Response: Thank you for raising this important point. We have added a comment in the discussion section (Lines 537-548). This human study did not include untreated patient controls or have sufficient power to determine if RA reduced mortality or length of stay in the hospital, but we did observe promising effects within 24 hours of RA treatment: subjective improvement in all subjects; increased SpO2; a trend toward lower CRP levels; rapidly reduced expression of CD69 on circulating iNKT cells; and a gradual reduction in circulating levels of 11 of 13 pro-inflammatory cytokines. The results are not adjusted for multiple statistical tests and some of the observed reductions in cytokine levels might be explained by regression to the mean. A larger placebo-controlled clinical trial is needed to confirm these findings, but they do suggest that RA will be effective for rapidly reducing cytokine production in inflamed COVID-19 patients.

As an aside, we ran the repeated measures models using the baseline value as a covariate and estimated the mean changes from baseline at 30minutes, 4 hours and 24 hours at the average of the baseline values. The results from these analyses were nearly identical to those found from the change point models presented in the paper. 

 Change scores Baseline as covariate

 Change from baseline SE Change from baseline* SE

IL-6 30m 0.178 0.344 0.176 0.311

 4.5h 0.204 0.344 0.203 0.311

 24h 0.422 0.344 0.421 0.311

MCP-1 30m 0.087 0.077 0.087 0.081

 4.5h 0.099 0.077 0.099 0.081

 24h 0.278 0.077 0.278 0.081

IL-8 30m -0.126 0.211 -0.125 0.643

 4.5h -0.026 0.211 -0.025 0.643

 24h -0.022 0.211 -0.021 0.643

IL12p40 30m -0.074 0.123 0.176 0.311

 4.5h -0.214 0.123 0.203 0.311

 24h -0.279 0.123 0.421 0.311

IL-7 30m -0.034 0.248 -0.034 0.261

 4.5h -0.249 0.248 -0.249 0.261

 24h -0.222 0.248 -0.222 0.261

IP-10 30m 0.084 0.126 0.084 0.121

 4.5h -0.275 0.126 -0.275 0.121

 24h -0.268 0.126 -0.268 0.121

IFN-g 30m -0.070 0.183 -0.070 0.175

 4.5h -0.313 0.183 -0.313 0.175

 24h -0.102 0.183 -0.103 0.175

MIP-1A 30m -0.108 0.409 -0.108 0.447

 4.5h -0.356 0.409 -0.356 0.447

 24h -0.647 0.409 -0.647 0.447

IL12p70 30m -0.047 0.158 -0.047 0.163

 4.5h -0.358 0.158 -0.358 0.163

 24h -0.271 0.158 -0.271 0.163

IL-17 30m 0.285 0.401 0.285 0.369

 4.5h -0.511 0.401 -0.511 0.369

 24h -0.723 0.401 -0.723 0.369

IL-1B 30m 0.017 0.366 0.016 0.353

 4.5h -0.685 0.366 -0.686 0.353

 24h -0.708 0.366 -0.709 0.353

IL-2 30m 0.035 0.432 -0.028 0.345

 4.5h -0.750 0.386 -0.752 0.308

 24h -0.795 0.386 -0.796 0.308

TNF-a 30m 0.049 0.355 0.049 0.365

 4.5h -0.754 0.355 -0.753 0.365

 24h -0.790 0.355 -0.790 0.365

*computed at the average value for the baseline. 

3. Please write under limitations that there is a risk of spurious results due to the number of statistical tests 

Response: We have added discussion regarding the risk of spurious results in the limitation section (lines 543-546) of the revised manuscript.

4. The two way anova under figure 2 is not mentioned under methods.

Response: Thank you. The two way anova under figure 2 was added to the Methods section (lines 248).

Reviewer #2: Overall this is an interesting paper describing the beneficial effects of a single infusion of Regadenoson, an A2A adenosine receptor agonist, on patients with COVID and murine data which further confirms the beneficial effects of this infusion. Moreover, the authors report that the principal antiinflammatory effects of regadenoson in this setting is mediated via inhibition of NKT cells. Overall, the clinical data remain quite preliminary and little can be said regarding efficacy of treatment in patients although the murine data is suggestive and prior clinical data using adenosine, a non-selective agonist, provides some evidence of efficacy in patients with COVID. There is one issue that should be corrected. The authors repeatedly note that adenosine cannot be given by IV infusion because of its effects on the heart (via A1 receptor stimulation). Adenosine is currently on the market for use as an infusion agent for pharmacologic stress testing so all of the references to A1 inhibitors and the contraindications to use of IV adenosine, etc should be eliminated.

Response: The reviewer is absolutely correct. We should have remembered Paracelsus of Ferrara who is famously credited with coining the phrase “the dose makes the poison”. As Paracelsus would have predicted the cardiovascular effects of adenosine are dose-dependent. In the revised manuscript we have revised our comment about adenosine simply to state that absence of any cardiovascular response to inhaled adenosine suggests that little accumulates in blood after inhalation, which is consistent with rapid uptake and metabolism into blood cells and ECs. We have removed the contraindications to use of IV adenosine in the revised manuscript.

---

## [Decision Letter · Decision Letter 1]

10 May 2023

PONE-D-22-17321R1Regadenoson for the treatment of COVID-19: a five case clinical series and mouse studiesPLOS ONE

Dear Dr. Lau,

Thank you for submitting your manuscript to PLOS ONE. I sincerely apologise for the unusually delayed review timeframe. Your revised manuscript was reviewed by one of the original reviewers, whose comments are appended below. The reviewer commented that their previous concerns were addressed but raised one minor point of concern that must be clarified before we can proceed. Therefore, we invite you to submit a revised version of the manuscript that addresses the points raised during the review process.

We look forward to receiving your revised manuscript.

Kind regards,

Emily Chenette

Editor in Chief

PLOS ONE

Journal Requirements:

Reviewers' comments:

Reviewer's Responses to Questions

**Comments to the Author**

1. If the authors have adequately addressed your comments raised in a previous round of review and you feel that this manuscript is now acceptable for publication, you may indicate that here to bypass the “Comments to the Author” section, enter your conflict of interest statement in the “Confidential to Editor” section, and submit your "Accept" recommendation.

Reviewer #1: (No Response)

2. Is the manuscript technically sound, and do the data support the conclusions?

Reviewer #1: Yes

3. Has the statistical analysis been performed appropriately and rigorously? 

Reviewer #1: I Don't Know

4. Have the authors made all data underlying the findings in their manuscript fully available?

Reviewer #1: No

5. Is the manuscript presented in an intelligible fashion and written in standard English?

Reviewer #1: Yes

6. Review Comments to the Author

Reviewer #1: All my comments have been adressed but regarding my question and your answer "The two way anova under figure 2 is not mentioned under methods. Response: Thank you. The two way anova under figure 2 was added to the Methods section (lines 248)."

A two way anova anlysis is not appropriate since it does not handle the repeated measures over time. Please double check which analysis that was performed or do a correct one. We can clearly see that there will be no effect on blood pressure but still the method for the statistical comparison should be correct.

7. PLOS authors have the option to publish the peer review history of their article (what does this mean?). If published, this will include your full peer review and any attached files.

Reviewer #1: No

---

## [Author Response · Author response to Decision Letter 1]

15 Jun 2023

Editor Q1: Is the manuscript now acceptable for publication?

Reviewer: no response

Author response: Please see response to Comment 3.

Editor Q2: Is the manuscript technically sound, and do the data support the conclusions?

Reviewer: yes

Author response: We appreciate that the reviewer found that the experiments had been conducted rigorously.

Editor Q3: Has the statistical analysis been performed appropriately and rigorously?

Reviewer: I Don't Know.

Author response: We have elaborated on statistical methods used in the figure legends.

Editor Q4: Have the authors made all data underlying the findings in their manuscript fully available?

Reviewer: No

Author response: To comply with then PLOS ONE data policy that requires authors to make all data underlying the findings described in their manuscript fully available, we have now added supplemental data tables including all of the data used for the figures.

Editor Q5: Is the manuscript presented in an intelligible fashion and written in standard English?

Reviewer: yes

Reviewer Q6: All my comments have been addressed but regarding my question and your answer "The two-way ANOVA under figure 2 is not mentioned under methods. A two-way ANOVA analysis is not appropriate since it does not handle the repeated measures over time. Please double check which analysis that was performed or do a correct one. We can clearly see that there will be no effect on blood pressure but still the method for the statistical comparison should be correct.

Author response: Thank you very much. Our statistician has reviewed and changed to repeated measures applied to changes from baseline. We have updated the manuscript appropriately with statistical explanations.

---

## [Decision Letter · Decision Letter 2]

7 Jul 2023

Regadenoson for the treatment of COVID-19: a five case clinical series and mouse studies

PONE-D-22-17321R2

Dear Dr. Lau,

We’re pleased to inform you that your manuscript has been judged scientifically suitable for publication and will be formally accepted for publication once it meets all outstanding technical requirements.

Again, I apologise for the unusually delayed review timeframe for this manuscript. I sincerely appreciate that you and your co-authors chose PLOS ONE as the venue for this study.

Kind regards,

Emily Chenette

Editor in Chief

PLOS ONE

Additional Editor Comments (optional):

Reviewers' comments:

Reviewer's Responses to Questions

**Comments to the Author**

1. If the authors have adequately addressed your comments raised in a previous round of review and you feel that this manuscript is now acceptable for publication, you may indicate that here to bypass the “Comments to the Author” section, enter your conflict of interest statement in the “Confidential to Editor” section, and submit your "Accept" recommendation.

Reviewer #1: All comments have been addressed

2. Is the manuscript technically sound, and do the data support the conclusions?

Reviewer #1: (No Response)

3. Has the statistical analysis been performed appropriately and rigorously? 

Reviewer #1: (No Response)

4. Have the authors made all data underlying the findings in their manuscript fully available?

Reviewer #1: (No Response)

5. Is the manuscript presented in an intelligible fashion and written in standard English?

Reviewer #1: (No Response)

6. Review Comments to the Author

Reviewer #1: (No Response)

7. PLOS authors have the option to publish the peer review history of their article (what does this mean?). If published, this will include your full peer review and any attached files.

Reviewer #1: No

---

## [Editor Report · Acceptance letter]

4 Aug 2023

PONE-D-22-17321R2 

Regadenoson for the treatment of COVID-19: a five case clinical series and mouse studies 

Dear Dr. Lau:

I'm pleased to inform you that your manuscript has been deemed suitable for publication in PLOS ONE. Congratulations! Your manuscript is now with our production department. 

Kind regards, 

on behalf of

Dr Emily Chenette 

Staff Editor

PLOS ONE